# Associations Between Clinical Inflammatory Risk Markers, Body Composition, Heart Rate Variability, and Accelerometer-Assessed Physical Activity in University Students with Overweight and Obesity

**DOI:** 10.3390/s25051510

**Published:** 2025-02-28

**Authors:** Sascha W. Hoffmann, Janis Schierbauer, Paul Zimmermann, Thomas Voit, Auguste Grothoff, Nadine Wachsmuth, Andreas Rössler, Tobias Niedrist, Helmut K. Lackner, Othmar Moser

**Affiliations:** 1Division of Theory and Practice of Sports and Fields of Physical Activity, BaySpo—Bayreuth Center of Sport Science, University of Bayreuth, 95440 Bayreuth, Germany; 2Division of Exercise Physiology and Metabolism, BaySpo—Bayreuth Center of Sport Science, University of Bayreuth, 95440 Bayreuth, Germany; janis.schierbauer@uni-bayreuth.de (J.S.); paul.zimmermann@uni-bayreuth.de (P.Z.); thomas.voit@uni-bayreuth.de (T.V.); auguste.grothoff@uni-bayreuth.de (A.G.); nadine.wachsmuth@uni-bayreuth.de (N.W.); 3Department of Physiology and Pathophysiology, Medical University of Graz, 8010 Graz, Austria; andreas.roessler@medunigraz.at (A.R.); helmut.lackner@medunigraz.at (H.K.L.); 4Clinical Institute of Medical and Chemical Laboratory Diagnostics, Medical University of Graz, 8010 Graz, Austria; tobias.niedrist@medunigraz.at; 5Interdisciplinary Metabolic Medicine Trials Unit, Department of Internal Medicine, Division of Endocrinology and Diabetology, Medical University of Graz, 8010 Graz, Austria

**Keywords:** sympathetic–parasympathetic regulation, objectively assessed physical activity, low-grade inflammation, biomarkers, sedentary time, health risk behavior, obesity, young adults

## Abstract

This cross-sectional study aimed to identify associations between clinical inflammatory risk markers, body composition, heart rate variability (HRV), and self-reported and objectively assessed physical activity (PA) in university students with overweight and obesity. Seventeen participants (eight females) completed a screening visit following a randomized controlled four-arm crossover trial period with 8 h of uninterrupted prolonged sitting, alternate sitting and standing, continuous standing, and continuous slow walking, respectively. Clinical inflammatory risk markers were obtained from venous blood samples, and PA was assessed using the International Physical Activity Questionnaire (IPAQ-SF) and ActiGraph wGT3X-BT accelerometers. HRV was recorded over 24 h using the Faros 180 Holter electrocardiogram (ECG). White blood cell (WBC) counts were significantly correlated with fat mass (FM; *p* = 0.03) and visceral adipose tissue (VAT; *p* = 0.04) and inversely correlated with moderate PA (*p* = 0.02). Light-intensity PA (LIPA) and moderate-to-vigorous PA (MVPA) were correlated with HRV parameters (*p* = 0.02), and LIPA was inversely correlated with interleukin-6 (*p* = 0.003) and c-reactive protein (*p* = 0.04) during different trial conditions. In university students with overweight and obesity, higher values of FM were negatively correlated with WBC count, and integrating LIPA and MVPA in the students’ daily life strengthened their sympathetic–parasympathetic regulation and positively mediated anti-inflammatory mechanisms.

## 1. Introduction

Regular physical activity (PA) is a key factor preventing non-communicable diseases (NCDs), such as obesity and type 2 diabetes mellitus, and leads to an improvement in both mental and physical health [1,2,3]. Nevertheless, the latest global estimates indicate that nearly 1.4 billion adults (27.5% of the world’s adult population) do not meet recommended levels of PA, and the World Health Organization (WHO) has classified physical inactivity as the fourth leading risk factor for global mortality [3,4].

Another leading cause of NCD is sedentary behavior (SB), which is commonly defined as any waking behavior with an energy expenditure of ≤1.5 metabolic equivalents of task (METs) while sitting, lying down, or in a reclining position [5,6]. High volumes of SB have become increasingly prevalent in modern societies due to changes in physical and social-environmental conditions and are associated with a greater risk for several major diseases and all-cause mortality [7,8]. The risk of all-cause mortality significantly increased above a threshold of 7.5 h/day of total sitting time, but there is evidence that an appropriate amount of moderate-to-vigorous physical activity (MVPA) of about 1 h/day appears to attenuate the increased all-cause mortality risk due to long periods of sitting [9]. Therefore, public health promotion efforts have to focus on increasing PA and reducing SB simultaneously [10].

Health risks associated with low PA values and high levels of SB have been documented across the human life span from school-aged children into adulthood [11,12,13]. In particular, university students have been identified as a population group at a high risk for high SB levels, and only 21 to 50% achieve the recommendations for PA [1,11,14,15,16]. Furthermore, university students appear to be affected by poor sleep quality and unhealthy lifestyle behaviors that may lead to excess body weight and obesity [16,17,18]. Since many health-related behaviors are developed during later adolescence and early adulthood, the transition from school to university, therefore, seems to be a crucial period for the development of future lifestyle and behavior patterns [1,15,19]. Sitting is omnipresent in the university context and anchored to typical university structures and procedures. University students are a specific sub-population at a high risk of accumulating high levels of SB due to typical student activities, such as attending lectures and seminars and studying [1,20]. Recent data indicate that university students spend on average 7.5 h sitting [15] with ~50% of students sitting at least 8 h per weekday [1], which is higher than the global average [4]. Moreover, accelerometer-based estimates of SB were about 2 h/day higher than self-reported data [15].

Health promotion among university students is not only important to positively influence students’ individual health but also could be sustainable and beneficial for the health of society from a health policy perspective. Students will be in fact the decision makers, executives, and parents of tomorrow [1,21]. For targeted interventions to promote PA and avoid SB, knowledge of the acute and long-term effects of different intensities of physical activity and sedentariness on health is essential, both for the unique group of university students and for the general population [15,22].

In this context, the replacement and interruption of uninterrupted prolonged SB even with light-intensity PA snacks (LIPAS) have recently been described to have positive effects on improving blood glucose levels and heart rate variability (HRV) parameters in university students with overweight and obesity [23]. Another study found out that SB and body fat mass (FM) had significant pro-inflammatory effects in a group of healthy participants between 18 and 55 years [24].

Furthermore, several investigations have shown that white blood cell (WBC) counts and subsets appear to be particularly sensitive to MVPA, and WBC counts seem to be increased in sedentary adults [25,26].

Besides adverse immune responses, autonomic cardiac modulation (ACM), commonly assessed using HRV, also contributes to the development of CVD [27]. It has been shown that reduced HRV is associated with an increased risk of all-cause mortality, whereby numerous factors can interfere with HRV, such as age, gender, overweight and obesity, and diabetes, as well as changes in PA and SB [28,29].

Due to the current scientific discourse on the health interactions between PA and SB, it is important to find out how the modifiable and health-influencing factors of PA and SB of students play a role in the university context and how these factors affect the individual individually or in combination [1,8,11,15,21]. Despite the large number of studies examining PA and SB in groups of university students [1,11,14,15,16,17,21], only a few studies have assessed PA and SB objectively [18,19,30,31], and none of them have investigated the interaction and relationship between clinical inflammatory risk markers and HRV parameters or objectively assessed PA and SB in university students with overweight and obesity. Therefore, the aim of the present study was to (1) represent clinical inflammatory risk markers and body composition parameters in students with overweight and obesity; and (2) identify associations between parameters of the sympathetic–parasympathetic balance and accelerometer-assessed PA in this vulnerable population group.

## 2. Materials and Methods

### 2.1. Study Design

The SED-ACT study with its primary outcomes [24] was designed as a single-center, prospective, randomized, controlled four-arm crossover trial investigating the influence of SB and PA on inflammatory and physiological processes in the human body. In December 2022, the study protocol was approved by the local ethics committee of the University of Bayreuth (Germany) (processing number 22-037), and this trial was registered at the German Clinical Trial Register (DRKS00031425). Before any trial-related activities were carried out, participants were informed about the study protocol and gave their written informed consent. Participants then were randomized to the order in which they should complete the four trial visits by a research associate that was not involved in the study using Research Randomizer^®^ 4.0 (Social Psychology Network, Lancaster, PA, USA) [32]. The participants took part in an initial screening visit and completed the following four 8 h trial visits in a random order with a 1-week washout period: (1) uninterrupted prolonged sitting (SIT), (2) alternate sitting and standing (SIT/STAND), (3) continuous standing (STAND), and (4) continuous slow walking at 1.6 km/h (WALK). Each participant completed the trial arms either at a height-adjustable office desk (Aeris^®^ Active Office, Haar, Germany; SIT, SIT/STAND, or STAND) or on a treadmill with a special shelf for books, tablets, and computers (LifeFitness Platinum Series, Life Fitness Europe, Unterschleißheim, Germany, WALK). The students were allowed to read, watch movies, work or study on the computer during the 8 h intervention period. The screening examination and the trial visits took place at our medical research laboratory at the Bayreuth Center of Sport Science (BaySpo) of the University of Bayreuth (Germany). For this analysis, we used unpublished cross-sectional data from the initial screening visit and each trial visit. A detailed description of the randomized controlled crossover study design, as well as the primary and secondary outcomes, have been published elsewhere [23,33]. This study was conducted in accordance with the principles of the declaration of Helsinki and Good Clinical Practice [34].

### 2.2. Eligibility Criteria

Participants were recruited via notices (digital and paper-based), the homepages of several organizational units of the University of Bayreuth, and if necessary, through social media. During a joint information and preparation meeting one week prior to the screening visit, the following eligibility criteria were defined and assessed by an investigator: participants with age ranging between 18 and 29 years (inclusive) and who were overweight or obese according to the WHO classification with a body mass index (BMI) ≥ 25.0 kg/m^2^. Participants were excluded if they were pregnant, had serious acute or chronic illnesses that precluded their participation in the study, had acute infection due to COVID-19, were simultaneously enrolled in a different study, or had (orthopaedical) restrictions that prevented them from being able to sit, stand, or walk for more than 8 h.

### 2.3. Screening Visit

The screening visit took place prior to the randomization. All participants were required to fast for at least 12 h and refrain from strenuous PA for at least 24 h. Participants were further not allowed to consume alcohol within 24 h before each visit. Body composition was determined using bioelectrical impedance analysis (BIA; Inbody 720, Inbody Co., Seoul, Republic of Korea), and height was measured manually (Seca 217, Seca, Hamburg, Germany). Standardized protocol for BIA was as follows: the participants remained seated for 10 min before undergoing a duplicate baseline measurement of their body composition according to the manufacturer’s instructions. Multifrequency technology is utilized in this body composition analyzer to separate intracellular and extracellular water by emitting a multitude of frequencies between 1 kHz and 1 MHz, reducing errors caused by individual variations in the distribution of total body water or its changes within a given period over time [35]. The InBody 720 is a reliable and valid tool for the assessment of whole-body composition measurement and has a high correlation to dual-energy X-ray absorptiometry (DEXA) [36,37,38,39]. The following parameters were analyzed during BIA: body weight, skeletal muscle mass (SMM), body fat mass (FM), and visceral adipose tissue (VAT) area. BMI was calculated as follows: weight (kg)/the square of height (m^2^). To clarify any abnormalities in the blood and to verify whether the participants’ glucose metabolism was impaired, we performed a standardized 75 g oral glucose tolerance test (OGTT; Glucoral^®^ 75 citron, Germania Pharmazeutika, Vienna, Austria) and complete blood count with a separate determination of glycated hemoglobin A1C (HbA_1c_) levels, which were assessed from a venous blood sample from the antecubital vein.

### 2.4. Blood Sampling

Blood samples were collected with a standard gauge cannula inserted into a subcutaneous vein. The cannula was occasionally flushed with sterile 0.9% saline solution to prevent blood clotting. At the screening visit, venous blood samples were obtained when the participants were fasting and dispensed evenly into lithium heparin tubes (BD Vacutainer^®^ SST^TM^ II Advance; BD Belliver Industrial Estate, Plymouth, UK). At each trial visit, before starting, blood samples were collected when the participants were fasting, 1 h after lunch [40], and at the end of the 8 h simulated condition. After resting a minimum of 30 min, the blood serum vacutainer was centrifuged at room temperature for 10 min at 3500× *g* (Rotanta/RP, Hettich AG, Bäch, Switzerland). Afterwards, serum was aliquoted and stored at −80 °C (ProfiLine Taurus, NationalLab, Mölln, Germany). Serum samples were analyzed in a single batch following the last trial visit. Using standardized assays by the same manufacturer and calibrated to international standards, all analyses were performed on a cobas 8000 analyzer (Roche Diagnostics GmbH, Mannheim, Germany).

### 2.5. Complete Blood Count and Inflammatory Markers

Blood samples during the screening visit were collected the morning after the participants had fasted. Data on erythrocyte count, platelet count, leucocyte count, hemoglobin, hematocrit Neutrophil granulocyte count, immature granulocyte count, eosinophil granulocyte count, basophilic granulocyte count, monocyte count, lymphocyte count, interleukin-6 (IL-6), c-reactive protein (CRP), and albumin were collected [33]. The systematic immune-inflammation index (SII) was calculated as follows: neutrophil count x platelet count/lymphocyte count [41,42]. The neutrophil-to-lymphocyte ratio (NLR) was calculated as follows: neutrophil granulocyte count/lymphocyte count [43,44]. The C-reactive protein-to-albumin ratio (CAR) was calculated as follows: C-reactive protein/albumin [45,46]. Venous blood samples to measure IL-6 and CRP during each trial visit were collected the morning after the participants had fasted before the start of the trial, 1 h after lunch, and at the end of the trial visit.

### 2.6. Physical Activity

During the screening visit, self-reported PA and sitting time were assessed using the German short form of the International Physical Activity Questionnaire (IPAQ-SF) [47]. The IPAQ is a reliable and valid assessment tool suitable for evaluating PA levels among university students [1,48,49]. Seven questions are included in the questionnaire, measuring the frequency (in number of days) and intensity (in minutes per day) of walking, moderate physical activities (MPAs), and vigorous physical activities (VPAs) in the past seven days. Furthermore, ST was also assessed by the amount of time spent sitting out of the number of hours per day [50]. MVPA in minutes per week was calculated by summarizing the amount of MPA and VPA.

During the trial period, PA was objectively assessed using ActiGraph wGT3X-BT accelerometers (ActiGraph LLC, Pensacola, FL, USA). Each participant wore an accelerometer on the hip of their dominant leg with an elastic waistband for seven consecutive days according to the manufacturer’s specifications and above the iliac crest [51], removing it only for water-based activities (e.g., showering and swimming) and during sleep, if necessary. In addition, participants kept an activity diary. Considering the participants’ answers about their self-reported daily activities in the PA log allowed us to crosscheck the data analyzed from the software to validate wear time more precisely [30].

### 2.7. Heart Rate Variability

HRV was recorded over 24 h using a one-channel Holter electrocardiogram (ECG) with a 500 Hz sampling rate (Faros 180, Bittium, Oulu, Finland) at each trial visit. The following standard HRV measures were evaluated in the time domain analysis: the standard deviation of the R-R-intervals (SDNN), the square root of the mean standard difference of successive R-R-intervals (RMSSD), and the ratio (LF/HF) between low frequency (LF) and high frequency (HF) [23,52,53]. HRV was assessed according to the guidelines from the European Society of Cardiology (ESC) and the North American Society of Pacing and Electrophysiology (NASPE) [54].

### 2.8. Data Processing and Statistical Analysis

An automatic report from Di Blasio et al. [50] was used to calculate the total amount of self-reported PA expressed in MET min/week at the time of the screening visit. By multiplying the minutes per day and days per week separately, the time spent performing moderate and vigorous activity and walking was calculated. Detailed description of the application and evaluation of the IPAQ and the method of scoring can be found elsewhere [1,47,49,50,55]. In accordance with the IPAQ-SF guidelines, questionnaires were considered invalid if any variable was missing or if the total sum of walking and moderate and vigorous activity, as well as the total sum of sitting per day, exceeded 960 min [55].

ActiLife software version 6.13.3 (ActiGraph LLC, Pensacola, FL, USA) was used to initialize the accelerometers and analyze the raw data. The devices were initialized to collect data at 30 Hz. The GT3X files were accumulated in ActiGraph count-based AGD files with an epoch length of 60 s and a non-wear time of 60 min based on the wear time algorithm from Choi et al. [56]. The outcome variables were overall PA counts per minute (cpm) (min/day). ST was defined as <100 cpm, time spent on light-intensity physical activity (LIPA) (min/day, 100–1951 cpm), time spent on MVPA (min/day, 1952–5724 cpm), and VPA (min/day, ≥5725 cpm) using the Freedson bout algorithm, which was originally validated on a comparable age group and is also recommended in public health research to represent a continuum in energy expenditure activities [57,58,59]. Calculation of daily MET amount was carried out with the Swartz Adult Overground and Lifestyle algorithm [60].

Regarding the HRV analyses, for frequency domain variables, a natural logarithmic transformation was applied due to skewed distributions. The continuous recorded HRV data were divided into three daytime periods to provide more clarity and to avoid artifacts during breakfast and lunch: between 9:00 a.m. and 11:00 a.m. in the morning, between 13:00 p.m. and 15:00 p.m. in the afternoon, and during the resting sleep phase of each participant at night. A resting state index was calculated using ECG data, 3D acceleration sensor data, and activity log data. The calculation for the restful sleep analysis has been described in detail elsewhere [61]. In addition, SDNN, RMSSD, LF, HF, and the LF/HF ratio were (1) calculated to a daily mean value for each condition for correlation analyses; and (2) summarized to a total mean value independently of the simulated condition for further regression analyses with the objectively assessed accelerometer PA variables, respectively.

During the trial period, inflammatory markers were collected after fasting before the trial, one hour after lunch, and at the end of the 8 h simulated trial visit. The mean value of IL-6 and CRP of each simulated condition was calculated for final regression analysis.

Data are presented as arithmetic means (95% CI). The Shapiro–Wilk normality test was used to assess the distribution of all data. HRV parameters obtained during trial period of the study sample for each condition were compared using analysis of variance (ANOVA) with Tukey correction. Pearson’s product moment, Spearman correlation, and simple linear regression analyses were performed to quantify the correlation between clinical inflammatory risk markers, body composition, HRV parameters, and physical activity. Statistical analysis was conducted using GraphPad Prism Version 8.0.2 (GraphPad Software, Inc., San Diego, CA, USA) and IBM SPSS Statistics 28 (IBM, Armonk, NY, USA). An a priori power analysis (n = 14) was performed for this study (G-Power, version 3.1.9.7, HHU-Düsseldorf, Germany) and is described elsewhere [23]. The level of significance was set to *p* < 0.05 (two-tailed).

## 3. Results

The participant flow diagram (Figure 1) shows that 19 out of the 47 recruited people consented to be screened and randomized, and 28 participants in total were withdrawn for not meeting the eligibility criteria. Of the randomized participants, two withdrew their participation due to personal reasons during the first intervention. The screening visit and all four trial visits were completed by 17 university students with overweight and obesity (eight females). The demographic and clinical inflammatory markers, body composition parameters, and physical activity values (IPAQ-SF) of the study sample are given in Table 1. The mean age was 23.4 (95% CI: 21.7, 25.0), and the mean study semester was 5.8 (95% CI: 3.5, 8.1). The basal anthropometry was a BMI of 29.7 kg/m^2^ (95% CI: 27.8, 31.6), whereas the adiposity markers showed a mean FM of 31.8% (95% CI: 27.6, 36.0) and a mean VAT area of 119.2 cm^2^ (95% CI: 101.7, 136.7). The complete blood count and differential blood count parameters showed no abnormalities.

The mean value for PA derived from the IPAQ-SF was 4250.9 MET min/week (95% CI: 2599.0, 5902.8), and the average self-reported sitting time was 8.9 h (7.4, 10.5) per weekday.

Table 2 shows the adjusted correlation values between clinical inflammatory risk markers, BC parameters, and PA at the time of the screening visit. Leucocyte count (r = −0.590, *p* = 0.014) and neutrophil granulocyte count (r = −0.564, *p* = 0.020) were negatively correlated with MPA. Regarding BC parameters, the VAT area and FM were positively correlated with subsets of granulocyte counts (r = 0.504 to 0.634, *p* ≤ 0.05), and IL-6 was positively correlated with body weight (r = 0.500, *p* = 0.043).

Figure 2 shows a simple linear regression analysis between the clinical inflammatory risk markers, body composition parameters, and physical activity at the time of the screening visit. Regarding the relationship between leucocyte count and MET min/week, the respective slope of the regression line indicates that a 100 MET min/week higher PA level was associated with a lower leucocyte count of −0.4 G/L, for example. An increase in the VAT area of 10 cm^2^ was associated with a higher neutrophil granulocyte count of 0.18 G/L, respectively. Finally, an increase in total body weight of 10 kg led to an increase in fasting IL-6 levels of 0.4 pg/mL.

### Results of the Trial Visits

Table 3 shows the descriptive characteristics of the accelerometer-assessed physical activity variables during the trial period. Overall, the participants spent 63.3% (95% CI: 61.6, 65.0) of their time during the week sedentary, 30.2% (95% CI: 28.4, 31.9) performing LIPA, and 6.5% (95% CI: 5.4, 7.6) performing MVPA. An overview of the heart rate variability parameters during the trial period is given in Table 4.

Table 5 shows the correlation values between the summarized mean HRV parameters and the inflammatory markers collected during each trial visit and the accelerometer-assessed physical activity values. Mainly, significant positive correlations were found for SDNN ms (r = 0.611, *p* = 0.015), RMSSD ms, (r = 0.697, *p* = 0.004), and HF m^2^ (r = 0.548, *p* = 0.035) with MVPA, whereas HF m^2^ (r = 0.600, *p* = 0.018) was also positively correlated with LIPA and negatively correlated with SLEEP (r = −0.646, *p* = 0.009).

Regarding the associations between the inflammatory markers and PA during the trial period, significant positive correlations were found between IL-6_SIT_ (r = 0.680, *p* = 0.007) and VPA, while CRP_STAND_ (r = −0.542, *p* = 0.040) was negatively correlated with LIPA. IL-6_WALK_ was also negatively correlated with the MET rate (r = −0.832, *p* < 0.001), LIPA (r = −0.745, *p* = 0.001), MVPA (r = −0.543, *p* = 0.039), and ST (r = −0.743, *p* = 0.002), whereas a positive correlation was found with SLEEP (r = 0.761, *p* = 0.001).

Figure 3 shows simple linear regression analyses between the heart rate variability parameters, inflammatory markers, and accelerometer-assessed physical activity variables during the trial period. Regarding the association between RMSSD and MVPA, the slope of the regression line indicates that a 30 min/week higher amount of MVPA was associated with a higher total RMSSD of 3.4 m^2^. Regarding the inflammatory markers IL-6 and CRP during the different conditions, an increase in LIPA of 10 h/week led to a decrease in IL-6_WALK_ of −1.4 pg/mL.

## 4. Discussion

University students are at a high risk for high levels of SB, low PA, and further unhealthy lifestyle behaviors and risk factors, like having overweight or obesity, poor mental health, and experiencing food insecurity [1,15,17,19,23,30,62]. To the best of our knowledge, this is the first study that (1) examined clinical blood samples of German university students; and (2) provided comprehensive information about the associations between parameters of the sympathetic–parasympathetic balance and accelerometer-assessed PA values in this vulnerable population group.

We found significant positive associations between specific WBC subsets and BMI, FM, and VAT area. This is in line with the literature [63], in which positive associations were also found between total WBC counts or differential counts and adiposity parameters (e.g., BMI, waist circumference (WC), total adipose tissue, subcutaneous adipose tissue, and visceral adipose tissue). Furthermore, self-reported MPA was negatively associated with total leucocyte count and neutrophil granulocyte count. As previously reported, MVPA seems to positively affect WBC counts in a dose-dependent manner in US adults [25]. An inverse association between cardiorespiratory fitness and total WBC count was also reported in a cohort of children and adolescents [26]. In contrast to other studies [43,64,65,66], although the correlation coefficients were between r = 0.226 and r = 0.453, we found no significant associations between several novel inflammatory markers, such as SII, NLR, and CAR, and body composition parameters.

The results of the trial period indicated that university students with overweight and obesity spend most of their time in a sedentary environment and performing LIPA. Only a small amount of 3.4% of MVPA min/week was recorded. Compared with a study from Finland, which investigated a cohort of young adults in the same age range, our study sample showed a lower amount of time spent performing daily LIPA and MVPA, respectively [67]. This is also true for a comparison with recently published data of university students from the US, in which the amounts of LIPA min/day and MVPA min/day were ~100 min/day and ~10 min/day higher than in our study, respectively [68]. Compared with results from another German student cohort [1], PA and SB levels were 1184 MET min/week and 1 h 29 min higher in our study.

Regarding the analysis between the HRV indices and objectively assessed PA, this research indicates that if university students spent a higher amount of time performing LIPA and MVPA in their daily life, the total RMSSD and total HF were positively altered during a simulated 8 h working and learning condition. Our results show significant effects for RMSSD in all conditions if the participants had a higher amount of MVPA min/week. The respective slope for an increase in total RMSSD per 30 min/week increase in MVPA was ~3.5 m^2^. This is in line with early research on the association between MVPA and HRV parameters, which also revealed positive associations of SDNN, RMSSD, and HF with MVPA in a cohort of healthy university students [69]. In contrast to other results [70], in our study, SB also showed significant positive associations with the total HF. This can probably be explained by the fact that our sample is very physically active and the MVPA was also positively associated with a high level of SB.

Our observations showed further a decreased release in CRP during a simulated 8 h workday with continuous standing, which was associated with a higher number of LIPA minutes per week. These results are congruent with those of Soares-Miranda et al. [69], in which they found that young adults with higher MVPA had the lowest CRP levels. Our study may extend this notion further to LIPA. Furthermore, the data showed that the lower the participants’ daily LIPA and MVPA levels, the more they benefited from the simulated slow walking condition regarding IL-6 release. A positive association was also observed between IL-6 during SIT and VPA min/week. It appears to be that a higher amount of VPA during the week led to an increased release of IL-6 during long uninterrupted prolonged sitting periods. Although a release of IL-6 is initially thought to have pro-inflammatory effects, a release in PA-induced IL-6 is in turn responsible for a release in circulating levels of anti-inflammatory cytokines such as IL-10 and IL-1ra, which further inhibits the production of tumor necrosis factor alpha (TNF alpha) and IL-1 beta [71,72,73] and may protect against the negative effects of uninterrupted prolonged sitting [74,75].

The strength of the present study is that it provides novel insights regarding the influence of several clinical inflammatory parameters and objectively assessed PA and SB levels on students’ health. To our knowledge, this is the first study that examined these parameters and their associations with each other in a sample of German university students with overweight and obesity. However, several limitations should be considered when interpreting our findings. Firstly, the findings of our study were obtained using a cross-sectional design. Larger studies, particularly with a longitudinal observational design, are needed to draw causal relationships and quantify our findings. Secondly, this study included a relatively small number of participants, and the obtained data and the conclusions of this study should be interpreted with caution for hypothesis generation and evidence verification. Thirdly, our study monitored activity levels, which may have introduced sampling bias since more active individuals were more likely to participate in this study. Fourthly, the study sample was restricted to a small group of university students, so the results cannot be generalized to other age and population groups or all university student populations. Although ActiGraph wGT3X-BT is widely used in diverse publications, hip-worn accelerometers only capture the amount of sedentary time, but not sedentary patterns (e.g., sitting and lying down). Furthermore, the use of different algorithms for the calculation of the PA amounts may lead to inaccurate and imprecise estimates.

### Practical Implications

The results from the present study indicate that university students may be at greater risk of high SB levels. In turn, objectively assessed PA data revealed that our participants were also very active and met the current PA recommendations [3,76]. Although there is a growing body of studies confirming high levels of SB in this population, to date, intervention studies reducing SB in the university context remain rare [1]. Our findings are important as they help to identify potential physiological consequences of inadequate PA levels in university students and point out that high values of FM and VAT may also negatively affect students’ health. Moreover, integrating regular bouts of LIPA seems to already have beneficial effects in terms of parasympathetic regulation for this young age group. An initial approach would therefore be to conduct an interventional longitudinal study in which students are regularly reminded or encouraged via wearables or apps to break up prolonged SB with LIPA in order to demonstrate the potential health-promoting effects of regular PA on students’ ACM, BC, and inflammatory status.

## 5. Conclusions

In summary, our study showed that high amounts of objectively assessed LIPA and MVPA strengthened the sympathetic–parasympathetic balance, and FM and VAT area are associated with circulating WBC counts. Furthermore, the data showed that the lower the participants’ general daily PA level, the more they benefited from incorporating LIPAS as often as possible into their day regarding IL-6 release. This also applies to circulating CRP during long standing periods over the day, which was reduced if the participants had higher levels of LIPA in their everyday life. Future research should therefore focus on the effectiveness of appropriate prevention and intervention programs in promoting daily LIPAS as often as possible into students’ everyday life.

## Figures and Tables

**Figure 1 sensors-25-01510-f001:**
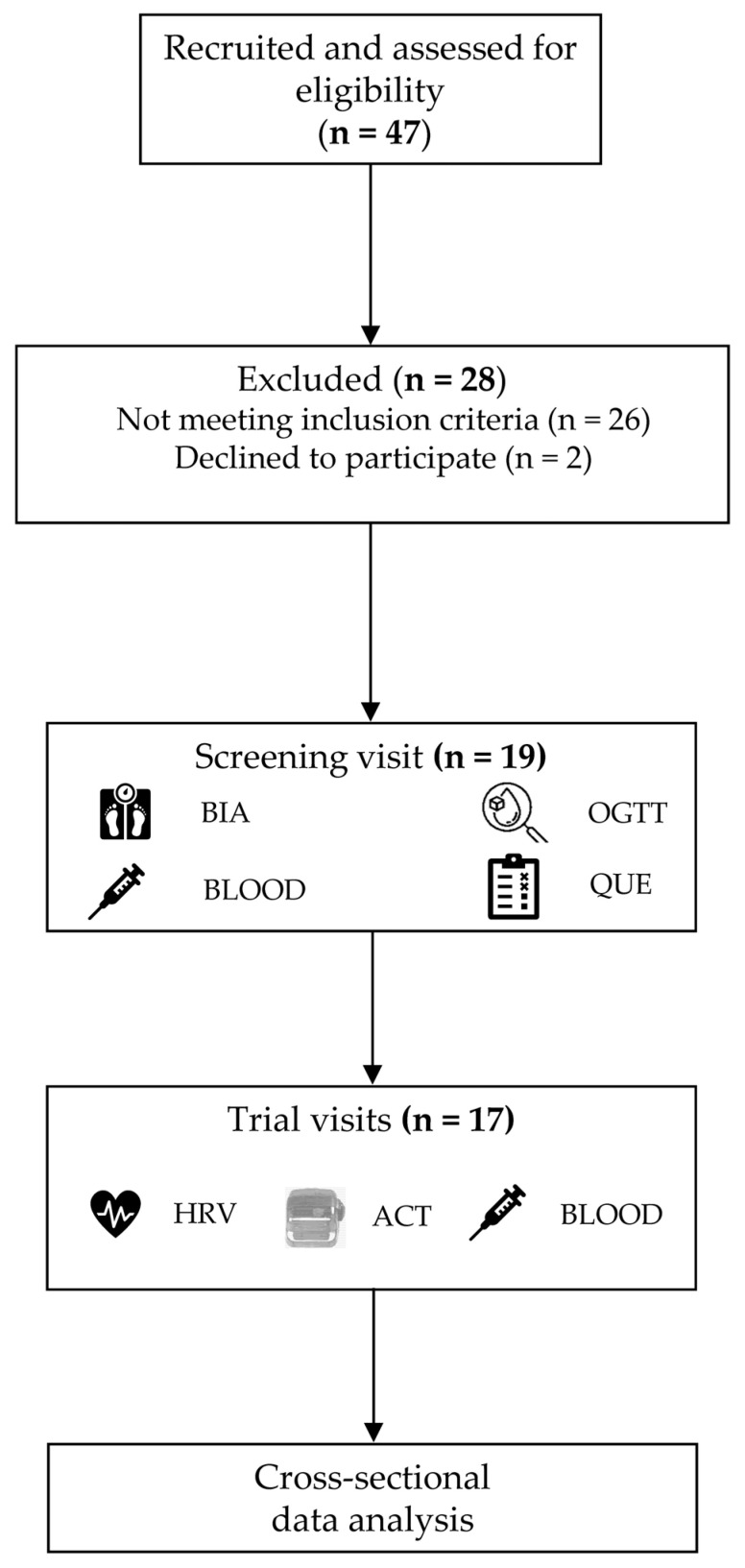
Participant flow chart. BIA, bioelectrical impedance analysis; OGTT, oral glucose tolerance test; BLOOD, venous blood sampling; QUE, questionnaire; HRV, heart rate variability; ACT, accelerometry with Actigraphy wGT3X-bt.

**Figure 2 sensors-25-01510-f002:**
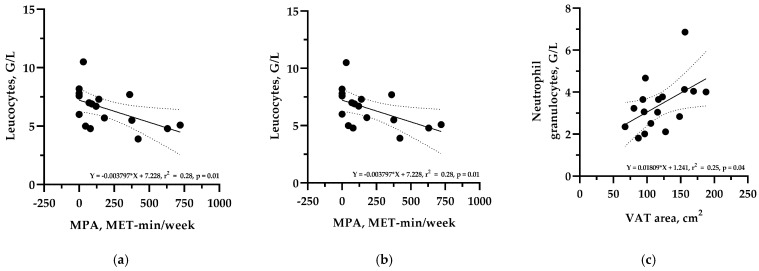
Simple linear regression analyses with the respective regression lines (bold lines) and the 95% CIs (dotted lines) between clinical inflammatory markers, body composition parameters, and physical activity variables at the time of the screening visit. (**a**) Leucocytes and moderate physical activity (MPA); (**b**) Neutrophil granulocyte count and moderate physical activity (MPA); (**c**) Neutrophil granulocyte count and visceral adipose tissue area (VAT); (**d**) Immature granulocyte count and body mass index (BMI); (**e**) Immature granulocyte count and fat mass (FM); (**f**) Eosinophil granulocyte count and fat mass (FM); (**g**) Eosinophil granulocyte count and visceral adipose tissue area (VAT); (**h**) Basophilic granulocyte count and fat mass (FM); (**i**) Basophilic granulocyte count and visceral adipose tissue area (VAT).

**Figure 3 sensors-25-01510-f003:**
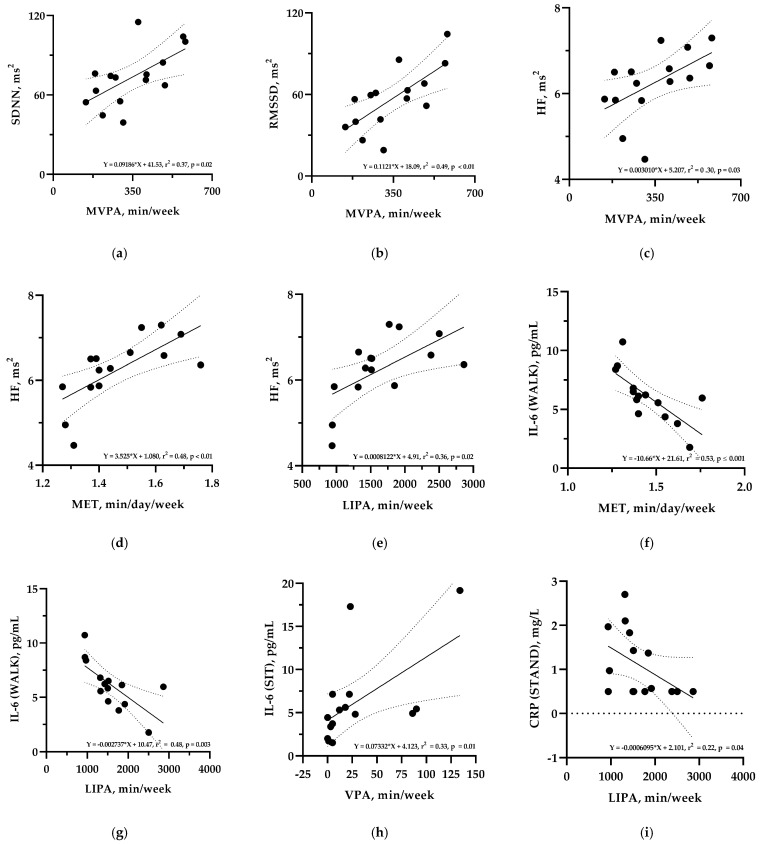
Simple linear regression analyses with the respective regression lines (bold lines) and the 95% CIs (dotted lines) between heart rate variability parameters, inflammatory markers, and accelerometer-assessed physical activity variables during the trial period. (**a**) Standard deviation of normal-to-normal beat (SDNN) and moderate-to-vigorous physical activity (MVPA); (**b**) Root mean square of successive differences (RMSSD) and moderate-to-vigorous physical activity (MVPA); (**c**) High frequency (HF) and moderate-to-vigorous physical activity (MVPA); (**d**) High frequency (HF) and metabolic equivalent of task (MET); (**e**) High frequency (HF) and light-intensity physical activity (LIPA); (**f**) Interleukin-6 (IL-6) during the WALK condition and metabolic equivalent of task (MET); (**g**) Interleukin-6 (IL-6) during the WALK condition and light-intensity physical activity (LIPA); (**h**) Interleukin-6 (IL-6) during the SIT condition and vigorous physical activity (VPA); (**i**) C-reactive protein (CRP) during the STAND condition and light-intensity physical activity (LIPA).

**Table 1 sensors-25-01510-t001:** Demographics, body composition parameters, physical activity values, and clinical inflammatory characteristics of the study participants during the screening visit.

Characteristics	Mean (95% CI, n = 17)	Characteristics	Mean (95% CI, n = 17)
**Demographics**		**Complete blood count**	
Females (n [%)]) *	8 (47.1)	Erythrocytes (T/L)	5.0 (4.8, 5.3)
Age (years)	23.4 (21.7, 25.0)	Platelets (G/L)	279.6 (247.4, 311.8)
Study semester	5.8 (3.5, 8.1)	Leucocytes (G/L)	6.5 (5.6, 7.3)
**Body Composition**		Hemoglobin (g/dL)	15.3 (14.5, 16.0)
Height (cm)	173.8 (167.5, 180.0)	Hematocrit (%)	43.5 (43.5, 47.8)
Weight (kg)	90.1 (80.7, 99.5)	Neutrophil granulocytes (G/L)	3.4 (2.8, 4.0)
BMI (kg/m^2^)	29.7 (27.8, 31.6)	Immature granulocytes (G/L)	0.02 (0.01, 0.03)
BMI categories		Eosinophils granulocytes (G/L)	0.2 (0.1, 0.2)
Overweight	10 (58.8)	Basophilic granulocytes (G/L)	0.04 (0.03, 0.05)
Obese	7 (41.2)	Monocytes (G/L)	0.6 (0.5, 0.7)
FM (%)	31.8 (27.6, 36.0)	Lymphocytes (G/L)	2.2 (1.9, 2.5)
SMM (%)	38.6 (36.0, 41.1)	**Inflammatory markers**	
VAT area (cm^2^)	119.2 (101.7, 136.7)	IL-6 (pg/mL)	3.2 (1.7, 4.8)
**Physical Activity, IPAQ**		CRP (mg/L)	2.6 (0.7, 4.5)
Walking (MET/min/week)	1168.6 (1743.3, 1105.9)	SII	479.8 (319.6, 639.9)
MPA (MET/min/week)	767.1 (299.9, 1234.2)	NLR	1.7 (1.2, 2.1)
VPA (MET/min/week)	2315.3 (1226.0, 3404.6)	CAR	0.5 (0.1, 0.9)
Total (MET/min/week)	4250.9 (2599.0, 5902.8)	Albumin (g/dL)	4.9 (4.7–5.0)
Sitting time (h/day)	8.9 (7.4, 10.5)	HbA_1c_ (%)	5.4 (5.2, 5.5)

* Proportion of females is given as an absolute number with relative value in %. Data are presented as means (95% CI), unless otherwise noted. IPAQ, International Physical Activity Questionnaire; BMI, body mass index, FM, fat mass; VAT area, visceral adipose tissue area; SMM, skeletal muscle mass; MPA, moderate physical activity; VPA, vigorous physical activity; MET, metabolic equivalent of task; min, minutes; IL-6, interleukin-6; CRP, C-reactive protein; SII, Systemic Immune-Inflammation Index; NLR, neutrophil-to-lymphocyte ratio; CAR, C-reactive-protein-to-albumin ratio.

**Table 2 sensors-25-01510-t002:** Correlation values between clinical inflammatory risk markers, body composition parameters, and physical activity derived from the screening visit.

Clinical Inflammatory Markers ^a^	Body Composition					Physical Activity ^b,c^
	BMI (kg/m^2^)	Body Weight (kg)	FM (kg)	VAT Area (cm^2^)	SMM (kg)	MPA (min/week)
**Complete blood count**						
Erythrocytes (T/L)	0.184	0.671 *	−0.069	0.247	**0.824 ****	0.134
Platelets (G/L)	0.193	−0.062	0.372	0.319	−0.266	−0.122
Leucocytes (G/L)	0.291	0.242	0.408	0.479	0.058	**−0.590 ***
Hemoglobin (g/dL)	−0.011	0.597 *	−0.234	0.134	**0.802 ****	0.202
Hematocrit (%)	−0.074	0.572 *	−0.258	0.109	**0.796 ****	0.226
Neutrophil granulocytes (G/L)	0.327	0.273	0.410	**0.504 ***	0.074	**−0.564 ***
Immature granulocytes (G/L)	**0.580 ***	0.413	**0.739 ****	**0.667 ***	0.110	−0.325
Eosinophil granulocytes (G/L)	**0.524 ***	0.393	**0.636 ****	**0.554 ***	0.150	−0.339
Basophilic granulocytes (G/L)	0.446	0.520	**0.515 ***	**0.634 ****	0.322	−0.211
Monocytes (G/L)	0.161	0.238	0.122	0.297	0.202	−0.431
Lymphocytes (G/L)	0.045	−0.054	0.150	0.084	−0.115	−0.157
**Inflammatory markers**						
IL-6 (pg/mL)	0.351	**0.500 ***	0.226	0.324	0.369	−0.038
CRP (mg/L)	0.250	0.416	0.345	0.368	0.365	0.035
SII (×10^9^ cells/L)	0.226	0.162	0.296	0.400	−0.083	−0.264
NLR	0.335	0.335	0.324	0.432	0.061	−0.164
CAR	0.330	0.300	0.394	0.453	0.005	−0.253

Correlation coefficients (Pearson or Spearman’s coefficient; adjusted for age and sex). Significant correlations with * *p* ≤ 0.05 and ** *p* ≤ 0.01. ^a^ n = 17. ^b^ Physical activity calculated with the International Physical Activity Questionnaire (IPAQ). ^c^ Only significant associations are displayed. IPAQ, International Physical Activity Questionnaire; BMI, body mass index; FM, fat mass; VAT area, visceral adipose tissue area; SMM, skeletal muscle mass; MPA, moderate physical activity; MET, metabolic equivalent of task; min, minutes; IL-6, interleukin-6; CRP, C-reactive protein; SII, Systemic Immune-Inflammation Index; NLR, neutrophil-to-lymphocyte ratio; CAR, C-reactive-protein-to-albumin ratio.

**Table 3 sensors-25-01510-t003:** Accelerometer-assessed physical activity during the trial period.

Characteristics	Mean (95% CI)
**Physical activity ^a^**	
SB (min/week)	3432.7 (2904.8, 3960.6)
SB (%)	63.3 (61.6, 65.0)
LIPA (min/week)	1632.3 (1352.5, 1912.2)
LIPA (%)	30.2 (28.4, 31.9)
MPA (min/week)	316.2 (254.9, 377.6)
MPA (%)	6.0 (5.0, 6.8)
MVPA (min/week)	346.1 (275.1, 344.2)
MVPA (%)	6.5 (5.4, 7.6)
VPA (min/week)	29.0 (8.7, 49.3)
VPA (%)	0.5 (0.2, 0.9)
METs (day/week)	1.5 (1.4, 1.6)
Sleep (h/day)	11.1 (9.2, 13.1)
Step counts (no/day)	7807.9 (6498.7, 9117.2)

^a^ Physical activity was assessed with the ActiGraph wGT3X-BT accelerometer. Data are presented as means (95% CI). ST, sitting time; LIPA, light-intensity physical activity; MPA, moderate physical activity; MVPA, moderate-to-vigorous physical activity; VPA, vigorous physical activity.

**Table 4 sensors-25-01510-t004:** Heart rate variability parameters according to the study condition during the trial period.

Characteristics Mean (95% CI)	SIT (1) ^a^	SIT/STAND (2) ^a^	STAND (3) ^a^	WALK (4) ^a^	*p*-Value *
**HRV parameters ^b^**					
SDNN (ms)	81.4 (68.2, 94.6) ^4^	80.9 (66.8, 94.9) ^4^	75.0 (62.1, 87.8) ^4^	55.9 (46.1, 65.7) ^1, 2, 3^	<0.001
RMSSD (ms)	64.9 (51.1, 78.7) ^3, 4^	62.5 (45.2, 79.8) ^4^	56.0 (44.5, 67.6) ^1, 4^	44.0 (33.7, 54.4) ^2, 3^	<0.001
LF (m^2^)	7.4 (7.1, 7.7) ^4^	7.5 (7.1, 7.8) ^4^	7.4 (7.0, 7.8) ^4^	6.8 (6.4, 7.2) ^1, 2, 3^	<0.001
HF (m^2)^	6.7 (6.3, 7.1) ^3, 4^	6.5 (6.1, 7.0) ^4^	6.2 (5.7, 6.7) ^4^	5.6 (5.1, 6.0) ^1, 2, 3^	<0.001
LF/HF-ratio (-)	0.7 (0.5, 1.0) ^3, 4^	0.9 (0.7, 1.2) ^3^	1.2 (0.9, 1.4) ^1, 2^	1.2 (1.0, 1.4) ^1^	<0.001

^a^ Mean differences in heart rate variability parameters between the conditions were calculated using one way analysis of variance (ANOVA) with Tukey correction. Significant differences between the conditions (SIT, 1; SIT-STAND, 2; STAND, 3; WALK, 4) are displayed with superscript numbers ^1–4^. Significant correlations with * *p* ≤ 0.05. ^b^ n = 15; individuals with missing data were excluded from the analyses. Data are presented as means (95% CI). HRV, heart rate variability; SDNN, standard deviation of normal-to-normal beat; RMSSD, root mean square of successive differences; HF, high frequency; LF, low frequency; SIT, uninterrupted prolonged sitting; STAND, continuous standing; SIT/STAND, alternate sitting and standing (2.5 h total standing time); WALK, continuous walking at 1.6 km/h.

**Table 5 sensors-25-01510-t005:** Correlation values between heart rate variability parameters, inflammatory markers, and accelerometer-assessed physical activity variables during the trial period.

	Physical Activity ^a^					
	MET day/week	LIPA min/week	MVPA min/week	VPA min/week	SB min/week	SLEEP min/week
**HRV parameters ^b^**						
SDNN, ms	**0.537 ***	0.336	**0.611 ***	**0.737 ****	0.407	−0.426
RMSSD, ms	**0.592 ***	0.363	**0.697 ****	**0.656 ****	0.448	−0.468
LF, m^2^	0.441	0.329	0.410	**0.636 ***	0.339	−0.363
HF, m^2^	**0.685 ****	**0.600 ***	**0.548 ***	0.467	**0.618 ***	**−0.646 ****
LF/HF ratio, -	**−0.642 ****	**−0.640 ***	−0.433	0.030	**−0.660 ****	**0.676 ****
**Inflammatory markers**						
IL-6 (SIT), pg/mL	0.195	0.091	0.391	**0.680 ****	−0.052	−0.061
CRP (SIT), mg/L	−0.043	−0.139	−0.034	−0.306	0.053	0.026
IL-6 (STAND), pg/mL	0.208	0.114	0.221	0.274	0.196	−0.179
CRP (STAND), mg/L	−0.321	**−0.542 ***	−0.015	−0.053	−0.226	0.290
IL-6 (SIT/STAND), pg/mL	0.023	0.086	0.043	−0.434	−0.050	0.011
CRP (SIT/STAND), mg/L	0.143	−0.095	0.369	−0.106	−0.084	−0.120
IL-6 (WALK), pg/mL	**−0.832 *****	**−0.745 ****	**−0.543 ***	−0.269	**−0.743 ****	**0.761 ****
CRP (WALK), mg/L	−0.193	−0.422	0.072	−0.165	−0.094	0.183

Significant correlations with * *p* ≤ 0.05, ** *p* ≤ 0.01, *** *p* ≤ 0.001. Correlation coefficients (Pearson or Spearman’s coefficient; adjusted for age, sex and body mass index, BMI). ^a^ Physical activity assessed with the ActiGraph wGT3x-BT accelerometer. ^b^ n = 15; individuals with missing data were excluded from the analyses. HRV, heart rate variability; PA, physical activity; MET, metabolic equivalent of task; LIPA, light-intensity physical activity; MVPA, moderate-to-vigorous physical activity; VPA, vigorous physical activity; SB, sedentary behavior; SDNN, standard deviation of normal-to-normal beat; RMSSD, root mean square of successive differences; HF, high frequency; LF, low frequency, IL-6, interleukin-6; CRP, C-reactive protein; SIT, uninterrupted prolonged sitting; STAND, continuous standing; SIT/STAND, alternate sitting and standing (2 h total standing time); WALK, continuous walking at 1.6 km/h.

## Data Availability

The raw data supporting the conclusions of this article will be made available by the authors on reasonable request.

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
