# Peer review of "Associations Between Clinical Inflammatory Risk Markers, Body Composition, Heart Rate Variability, and Accelerometer-Assessed Physical Activity in University Students with Overweight and Obesity"

_sensors, 2025, doi:10.3390/s25051510_

Round 1
Reviewer 1 Report
Comments and Suggestions for Authors
This paper looks at how physical activity and sitting around are linked to things like inflammation, body fat, and heart rate in overweight and obese college students. It's a good topic, and they could have some really interesting findings here, but the paper needs some serious work before it's ready for prime time. They've got a ways to go to make it solid and clear. They need to really focus on accurately describing what they did, beefing up the methods section, and making the results and discussion easier to follow. If they can do that, this study could actually teach us something important about how activity levels affect these students' health. Major revisions are recommended, with a particular emphasis on accurately reflecting the study design, strengthening the methodological details, and improving the clarity of the results presentation and discussion. Addressing these issues will enhance the rigor and impact of the study.
Article name:
They call it a randomized controlled crossover trial, but it really isn't. They had students do different things like sitting, standing, and walking, but they didn't use those activities as interventions to change students' weight or anything like that. They seem to just have used those different activities to get a range of activity and sedentary levels in their data so they could see how those levels correlated with things like inflammation markers. They're looking at associations, not effects. That's a big difference. They need to rewrite the title, intro, and pretty much the whole framing to make it clear this is an observational, cross-sectional study. A better title might be something like: "What's the Connection? Physical Activity, Body Composition, Inflammation, and Heart Rate in Overweight and Obese Students."
About that intro: It's way too long. They've got 41 references in there! It's hard to figure out what the main point is. They need to cut it down and clearly explain why this study is needed and what they expected to find.
The methods section needs some work, too:
Body Composition: They used a body fat analyzer (Inbody 720), but they don't explain why they chose that particular one. Is it good for this population? They need to show that it's reliable.
Simulated Work: They talk about a "simulated work environment," but it's vague. What did the students actually do in each condition? What equipment did they use? What did the room look like?
Accelerometers: They used ActiGraphs, which is good, but where exactly on the hip did they put them? And what did they do with the activity log beyond just "checking" it against the software? There's not enough detail here.
Cut-Points: They give the cut-points for sedentary time, light, moderate, and vigorous activity, but they don't explain why they picked those. They need to back that up with some evidence.
The results section could be clearer:
Figures 2 and 3: They show some relationships, but where are the equations and R-squared values? That would help us see how strong those relationships are.
Table 5: It's a long table. Maybe they could break it down or use a heatmap or something. It is hard to read.
Some other things:
Discussion: What are the implications? What should other researchers do next?
References: They need to make sure all the references are formatted correctly.
Reviewer 2 Report
Comments and Suggestions for Authors
Dear Corresponding Author, thank you for submiting your manuscript to Sensors journal and congratulations on your work.
Brief Summary: The study examinated the associations between clinical-inflammatory risk markers, body composition, heart rate variability (HRV) and physical activity assessed by accelerometer in university students with overweight and obesity. A four-arm randomized controlled crossover trial was conducted on 17 participants, comparing 8 hours of: uninterrupted prolonged sitting, sitting/standing alternation, continuous standing position and continuous slow walking. Blood samples were collected for inflammatory markers, body composition analysis via BIA, physical activity monitoring with ActiGraph accelerometers and 24h HRV monitoring with Faros 180 Holter ECG.
General Comments: The research addresses a relevant and current topic with results that provide new evidence on the relationships between physical activity and physiological parameters. However, in my opinion some aspects could be improved because they did not convince me, for exemple:
- The introduction could benefit from additional references on sedentary behavior in university students, although being a young population it is well established in literature that the physical activity deficit is considerable, it would be appropriate to identify this aspect.
- The discussion of study limitations could be expanded, particularly regarding the sample size which I do not consider particularly large to reach a generalization
- Practical implications of the results could be added because the conclusion did not appear particularly concrete to me, at least as my personal oppinion
Specific Comments:
Introduction:
- Lines 103-106: add more recent references on sedentary behavior of university students as anticipated before. It seems forced to say "little is known about the PA and ST behavior within the collective of university students and its effects on students' health" because this topic has been discussed in literature for some time.
- Line 160: BIA also needs to follow a standardized protocol that includes not only alcohol and meals, what protocol did the athletes follow before the test?
Methods:
- line 177-179: better specify the standardization criteria for the pre-test evening meal because it can lead to bias, it's not clear what "the same meal" means, at least I didn't understand it
Results:
- Table 2: improve readability by adding horizontal lines between sections
Discussion:
- Page 14, lines 450-460: expand comparison with existing literature
- Page 15, lines 531-537: deepen study limitations, it seems quite reductive to say that only "small sample size" is a clear limitation.
- Add a paragraph on practical implications of results for university students
In conclusion, the manuscript is of good quality and deserves publication after minor revision of the aspects indicated above. I look forward to reading a final version to give a last decision
Round 2
Reviewer 1 Report
Comments and Suggestions for Authors I have no further suggestions at this time.